# A deep learning–based algorithm for tall cell detection in papillary thyroid carcinoma

Sebastian Stenman [1,2,3]*, Nina Linder[1,4], Mikael Lundin[1], Caj Haglund[3,5☉], Johanna Arola[2☉], Johan Lundin[1,6,7☉]

**1** Institute for Molecular Medicine Finland – FIMM, University of Helsinki, Helsinki, Finland, **2** HUSLAB Pathology Department, Helsinki University Hospital, Helsinki, Finland, **3** Department of Surgery, Helsinki University Hospital, Helsinki, Finland, **4** Department of Women's and Children's Health, International Maternal and Child Health at Uppsala University, Uppsala, Sweden, **5** Research Programs Unit, Translational Cancer Medicine, University of Helsinki, Helsinki, Finland, **6** Department of Global Public Health, Karolinska Institutet, Stockholm, Sweden, **7** iCAN Digital Precision Cancer Medicine Flagship Helsinki, Helsinki, Finland

☉ These authors contributed equally to this work.
* sebastian.stenman@helsinki.fi

**Data Availability Statement:** The images used to validate the deep learning based algorithm and the results of the trained algorithm can be accessed by the reader via the following link (https://cloud.aiforia.com/Public/LundinLab/eom7FCyANBZ8boHHo1hXJtsXIjcbW5M6Ujmd2bnUIx80).

## Abstract

### Introduction

According to the World Health Organization, the tall cell variant (TCV) is an aggressive subtype of papillary thyroid carcinoma (PTC) comprising at least 30% epithelial cells two to three times as tall as they are wide. In practice, applying this definition is difficult causing substantial interobserver variability. We aimed to train a deep learning algorithm to detect and quantify the proportion of tall cells (TCs) in PTC.

### Methods

We trained the deep learning algorithm using supervised learning, testing it on an independent dataset, and further validating it on an independent set of 90 PTC samples from patients treated at the Hospital District of Helsinki and Uusimaa between 2003 and 2013. We compared the algorithm-based TC percentage to the independent scoring by a human investigator and how those scorings associated with disease outcomes. Additionally, we assessed the TC score in 71 local and distant tumor relapse samples from patients with aggressive disease.

### Results

In the test set, the deep learning algorithm detected TCs with a sensitivity of 93.7% and a specificity of 94.5%, whereas the sensitivity fell to 90.9% and specificity to 94.1% for non-TC areas. In the validation set, the deep learning algorithm TC scores correlated with a diminished relapse-free survival using cutoff points of 10% (p = 0.044), 20% (p < 0.01), and 30% (p = 0.036). The visually assessed TC score did not statistically significantly predict survival at any of the analyzed cutoff points. We observed no statistically significant difference in the TC score between primary tumors and relapse tumors determined by the deep learning algorithm or visually.

Clinical data of the cases included in the validation series have been collected from Helsinki and Uudenmaan sairaanhoitopiiri (HUS) and encrypted clinical data can be made available upon request. Data from the Cancer Genome Atlas (www.cancer. gov/about-nci/organization/ccg/research/ structural-genomics/tcga) used in this study is readily available and can be accessed by the general public.

**Funding:** This study has received funding from; Syöpäsäätiö Cancer Foundation Finland funded J. A., S.S., (www.syopasaatio.fi), Finska Läkaresällskapet funded S.S., C.H., J.L. (www.fls. fi), K. Albin Johanssons Foundation funded S.S. (www.foundationweb.net/johansson/), Sigrid Juséliuksen Foundation funded S.S., C.H., J.L. (www.sigridjuselius.fi), Medicinska understödsföreningen Liv och Hälsa funded N.L., C.H., J.A., and J.L. (www.livochhalsa.fi), iCAN Digital Precision Medicine Flagship funded N.L., and J.L. (www.ican.fi), and HiLIFE Helsinki Institute of Life Sciences funded N.L., and J.L., (www2. helsinki.fi/en/helsinki-institute-of-life-science). The funders had no role in study design, data collection and analysis, decision to publish or preparation of the manuscript in any way.

**Competing interests:** Johan Lundin and Mikael Lundin are founders and co-owners of Aiforia Technologies Oy, Helsinki, Finland.

## Conclusions

We present a novel deep learning–based algorithm to detect tall cells, showing that a high deep learning–based TC score represents a statistically significant predictor of less favorable relapse-free survival in PTC.

## 1. Introduction

Papillary thyroid carcinoma (PTC) has the most favorable outcome among all thyroid carcinomas, especially in young patients [1–3]. However, the tall cell variant (TCV) of PTC correlates with a more aggressive disease and less favorable outcomes [4–6]. TCV is associated with a greater risk of recurrence and further extra-thyroidal extensions [6–8]. The World Health Organization (WHO) defines a TCV as a PTC containing at least 30% epithelial cells that are two to three times as tall as they are wide with an abundant eosinophilic cytoplasm [9]. This threshold percentage, however, remains debated. A TC score as low as 10% was previously proposed as correlating with an adverse outcome [5, 10]. Yet, in other studies higher thresholds for the TC score were reportedly needed to identify an adverse outcome, such as thresholds of 30% [6], 50% [7], and 70% [11]. The debate regarding the TCV threshold causes confusion among pathologists [12, 13].

The visual assessment of the TC composition through conventional microscopy remains a subjective and time-consuming task, leading to high inter- and intra-observer variability demonstrated through the reevaluation of tissue samples [3, 14]. Whole-slide imaging (WSI) and computational methods allow for the quantitative analysis of increasingly complex morphological patterns. Deep learning–based algorithms have been used for a wide range of tasks, from the detection of cell nuclei [15], mitoses [16], and tumor-infiltrating immune cells [17, 18], to more complex spatial pattern recognition within tumors [19, 20] and tumor grades [21]. These methods can help tackle inter- and intra-observer variability and subjectivity when analyzing tissue samples.

In this study, we evaluated the feasibility of a deep learning–based tissue segmentation method to assess TCs in thyroid carcinomas. The process includes the segmentation of tumor tissue as the first step, followed by the segmentation of the epithelium into TC and non-TC areas and, thus, a TC percentage—thus, a TC score, can be calculated. Specifically, we aimed to evaluate if deep learning (DL) methods can be applied to the quantification of the TC composition and how that correlates with TC scores assessed by human observers. By using this novel TCV algorithm, we also studied the association between the TC score and PTC outcomes at various cutoff points in a selected cohort of PTC patients. As a secondary aim of the study, we analyzed 71 PTC relapse samples to study how the TC composition correlates to the morphology in the primary tumors.

## 2. Methods

### 2.1. Patient series

**2.1.1. Training series.** The DL algorithm was trained using 100 whole-slide images (WSIs) of hematoxylin and eosin (H&E) stained tissue samples originating from 100 separate patients with PTC. Among these, 70 WSIs originated from a PTC cohort treated from 1973 to 1996 at the Helsinki University Hospital [2, 11]. To broaden the training dataset and improve the generalizability of the trained algorithm, 30 additional PTC WSIs were added to the

training series. These 30 WSIs were downloaded from The Cancer Genome Atlas (TCGA) [22] (Fig 1), a vast publicly available database of information including genomic data and histological WSIs of 33 different cancer types.

**2.1.2. Validation series.** We validated the trained DL algorithm on an independent case–control cohort comprising 90 PTC patients treated at the Hospital District of Helsinki and Uusimaa (HUS) between 01/01/2003 and 12/31/2013. The follow-up time ended in 12/31/2018 and therefore allowed all included patients to have a follow-up of at least 5 years. Patients with an adverse outcome (n = 34) were defined as PTC cases with at least two recurrences (histological confirmation or serum thyroglobulin elevation during follow-up), distant metastases, or patients who died from PTC. These adverse outcome patients were matched with 1 to 2 controls (n = 56) according to age (within 10 years), gender, and tumor stage (T stage). Microcarcinomas (<1 cm in diameter) were excluded from this cohort (Fig 1). Formalin-fixed paraffin-embedded tissue blocks for all patients treated at HUS were retrieved and simultaneously assessed by two researchers (SS and JA) using a multiview microscope. Based on these slides, the most representative tissue block for each patient was selected. New tissue sections of these representative blocks were cut and stained with H&E following standard procedures. The H&E-stained slides were then digitized with a scanner (Pannoramic 250 FLASH 3DHISTECH Ltd., Budapest, Hungary) equipped with a plan-apochromat at objective 20x (NA 0.8), a CMOS camera (Adimec Q-12A-160Fc, Eindhoven, The Netherlands) with a pixel size of 0.2 μm/pixels and a 1.6 adapter. Following digitization of the slides, they were compressed into a wavelet format (Enhanced Compressed Wavelet, ECW, ER Mapper, Intergraph, Atlanta, GA, USA) with a compression ratio of 1:9 and imported to an image management platform (Aiforia Create, Aiforia Technologies Oy, Helsinki, Finland). Patient follow-up time ranged from 2.1 to 15.8 years (median 10.1 years). The median age at diagnosis was 41.0 years (standard deviation [SD] ± 16.2) in the adverse outcome group, 41.5 years (SD ± 14.1) in the control group, and 41 years (SD ± 14.9) for the entire cohort. During follow-up, all 34 patients in the adverse outcome group experienced disease relapse compared with 14 patients in the control group. The median relapse–free survival (RFS) was 0.8 years (SD ± 1.2) in the adverse outcome group and 8.6 years (SD ± 4.5) in the control group. In the adverse outcome group, 2 patients had distant metastases at the time of primary diagnosis, 34 patients had relapses, and 7 patients were diagnosed with distant metastases during follow-up, 2 patients died of PTC, and 6 patients died from other causes (Table 1).

**2.1.3. Relapse series.** All available histologically evaluated tissue samples (n = 71) from relapses in patients in the adverse outcome group (n = 34) were collected, visually assessed, and a representative FFPE tissue sample was selected for each of the relapses. New fresh tissue slices were cut from these representative FFPE tissue samples, stained with H&E, and digitized according to the same procedure protocol as previously described.

## 2.2. Training of the deep learning algorithm

The DL-based model trained to assess the TC score consists of two algorithms run in sequence. The first algorithm was trained to segment the tumor tissue (Fig 2). The second algorithm was trained to segment the tumor tissue into TC and non-TC areas. Both algorithms were trained and tested on manual annotations of the regions of interest within the 100 PTC WSIs originating from 100 separate patients. The manually annotated regions of interest were 2,970 in total and were carried out by one of the researchers (SS). Among the manual annotations, 90% (n = 2,674) were used for training and the remaining 10% (n = 296) were used as a holdout test set for the assessment of the performance of the algorithm (Fig 1). To improve the generalizability of the model, image augmentations by perturbation of the training data were utilized.

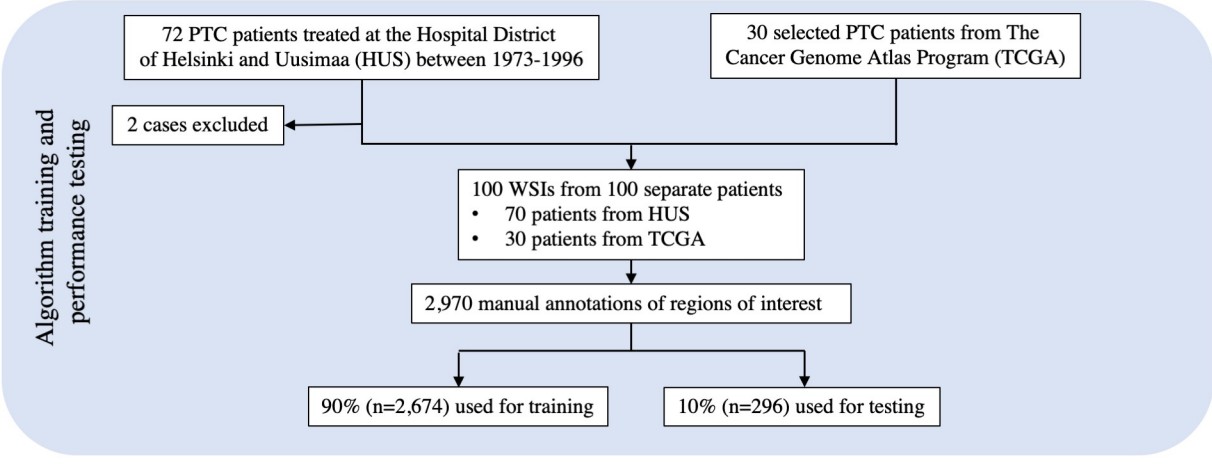

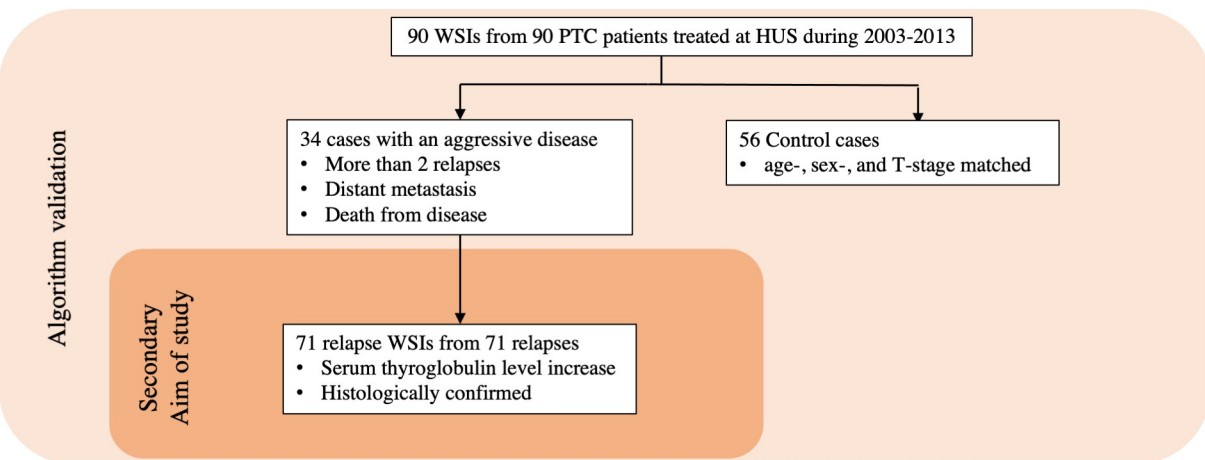

**Fig 1. CONSORT flow diagram of the datasets used in the study.** Papillary thyroid carcinoma is abbreviated as PTC.

The augmentations used to train the tumor tissue segmentation algorithm consisted of the rotation (0–360˚), variation of scale (±10%), shear distortion (±10%), aspect ratio (±10%), contrast (±10%), white balance (±10%), and luminance (±10%). The predetermined feature size was 500 μm. When training the TC segmentation algorithm, the augmentations consisted of the rotation (0–360˚), variation of scale (±5%), contrast (±20%), white balance (±20%), and luminance (±20%) with a predetermined feature size of 40 μm. Training of the model was continued until no decrease in the area error was detected for 500 iterations (training epochs) or until a total of 10 000 iterations was reached. The final model was trained with 9,521 completed iterations.

## 2.3. Tall cell analysis

The TC score for the representative WSIs of the primary tumors as well as the relapse tumors was visually assessed by one of the researchers (SS) who was blinded to the clinical characteristics of the patients. WSIs were grouped by the visual TC score into three groups of 0–9%, 10–29%, and ≥30% TCs (Table 2). The TC scores were also grouped into 10 percentage groups: 0–9%, 10–19%, 20–29%, 30–39%, 40–49%, 50–59%, 60–69%, 70–79%, 80–89%, and 90–100% (S1 Table in S1 File). The same WSIs were analyzed separately by the trained DL-based

**Table 1. Characteristics of papillary thyroid carcinoma (PTC) patient cohort.** The cohort comprised 34 patients with an adverse outcome and 56 age-, sex-, and tumor stage–matched control PTC patients.

| Patient characteristics | Adverse outcome (n = 34) | Control (n = 56) |
|---|---|---|
| Female | 23 (68%) | 41 (73%) |
| Male | 11 (32%) | 15 (27%) |
| Nodal metastases | 25 (74%) | 28 (50%) |
| Primary distant metastases | 2 (6%) | 0 (0%) |
| Relapse | 34 (100%) | 14 (25%) |
| Distant metastases during follow-up | 7 (21%) | 0 (0%) |
| Died during follow-up | 6 (18%) | 2 (4%) |
| Died of PTC | 2 (6%) | 0 (0%) |
| Primary RAI | 33 (97%) | 55 (98%) |
| Median age at diagnosis (in years) | 41.0 | 41.5 |
| Median follow-up time (in years) | 10.4 | 9.7 |
| Median relapse-free survival (in years) | 0.8 | 8.6 |
| Stage of tumor* | | |
| • T1 | 5 | 10 |
| • T2 | 10 | 19 |
| • T3 | 17 | 25 |
| • T4 | 2 | 2 |
| RAI times (mean) | 3.4 | 3.0 |
| Algorithm TC score (median) | 32% | 23% |
| Visual TC score group (median) | 30–39% | 0–9% |

Figures represent number of patients unless otherwise stated

RAI, radioiodine ablation

*T classification and TNM stage according to the TNM classification, seventh edition of the American Joint Committee on Cancer staging of papillary thyroid cancer.

algorithm. The TC scores of the DL algorithm were reported using a continuous scale, and also grouped in 10% incremental percentage groups as well as in three groups of 0–9%, 10–29%, and ≥30% TC in order to compare the visually assessed TC scores. The images included in the study can be viewed via the following URL: https://tinyurl.com/9pbyuuxm.

## 2.4. Statistical analysis

All statistical analyses were calculated using a general-purpose statistical software package (Stata 16.1 for Mac, Stata Corp., College Station, TX, USA). The performance of the DL algorithm was evaluated by calculating the sensitivity and specificity based on the independent test set. The F1 score of the independent test set was also assessed as a harmonic mean of the sensitivity (recall) and positive predictive value (precision). The statistical distribution of the samples according to their TC score were analyzed using the Mann–Whitney U test. The Fisher's exact test was used to statistically assess the differences between groups for nominal variables and the Cochran–Armitage tests for trends between ordinal variables. Agreement between the researcher's and the algorithm's TC score was tested with weighted kappa statistics with linear weights. The Kaplan–Meier method with the log-rank test and the Cox proportional hazard regression model were calculated for the survival analyses. Relapse-free survival (RFS) was defined as the time between the primary operation until relapse or end of follow-up. Overall survival (OS) was defined as the time between the primary operation and death from any

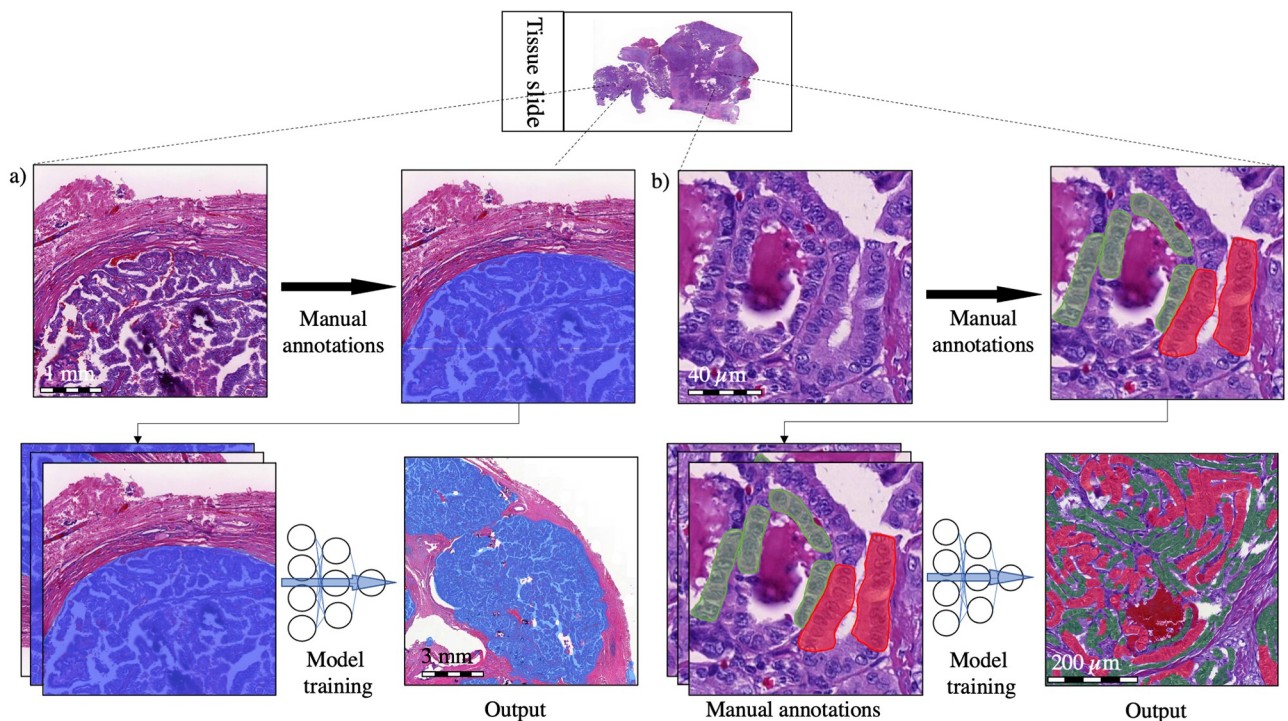

**Fig 2. The convolutional neural network model consisted of two algorithms.** a) A segmentation algorithm detects the tumor tissue (blue), which is fed as an input to a b) tall cell (TC) segmentation algorithm trained to detect TC epithelial regions (red) as well as non-TC epithelial regions (green). Finally, the TC score of the total epithelial area was calculated.

cause. We used tall cell score as the primary exposure in survival analyses. Spearman's rank correlation coefficient evaluated correlations between variables. We considered $p < 0.05$ as statistically significant using two-tailed tests.

## 2.5. Ethical statement

We used retrospective samples that were routinely collected. This study complies with the Declaration of Helsinki and was approved by the Surgical Ethics Committee of the Helsinki University Hospital (DNo. HUS 226/E6/06, extension TMK02 §66 17.04.2013). The National Supervisory Authority of Health and Welfare granted us permission to use the tissue samples without requiring individual informed consent in this retrospective study (Valvira DNo. 10041/06.01.03.01/2012).

**Table 2. The interrater agreement between the visually assessed tall cell (TC) percentage and the algorithm's TC area–based percentage in the primary tumors.** An interrater agreement analysis was performed and yielded a weighted kappa value of 0.31 (SD ± 0.068).

| | | Algorithm TC score assessment | | | Total |
|---|---|---|---|---|---|
| | | <10% | 10–29% | ≥30% | |
| Visual TC score assessment | <10% | 6 | 27 | 7 | 40 |
| | 10–29% | 1 | 7 | 8 | 16 |
| | ≥30% | 1 | 7 | 26 | 34 |
| | Total | 8 | 41 | 41 | 90 |

## 3. Results

### 3.1 The deep learning algorithm

In the test set containing 296 manual annotations, the algorithm detecting tumor tissue reached a positive predictive value (PPV; precision) of 99%, a sensitivity of 99%, and an F1 score of 99%. The subsequent TC segmentation algorithm detected TC regions with a PPV of 95%, sensitivity of 94%, and F1 score of 94%. Non-TC regions were detected with a PPV of 94%, sensitivity of 91%, and F1 score of 92% (Fig 3). In the validation dataset, the DL algorithm detected the median TC percentage area—that is, a TC score of 22.8% (SD ± 13.0%) in the control group and 31.6% (SD ± 11.8%) in the group with an adverse outcome. The algorithm results can be visually assessed via the following link: https://tinyurl.com/bde78hby.

### 3.2. Agreement between visual and algorithm-based tall cell scores

The interrater agreement between the human and algorithm TC scoring when analyzed according to three groups (<10%, 10–29%, and ≥30%) yielded a weighted kappa value of 0.31 (SD ± 0.068), which translates to a fair agreement (Table 2). The interrater agreement was also calculated using all ten TC score groups, yielding a weighted kappa value of 0.36 (SD ± 0.058; S1 Table in S1 File).

### 3.3. Algorithm-based tall cell score and survival

Overall, a higher algorithm-based TC score correlated with an adverse outcome (p = 0.005). When studying the TC score in 10% increments for the correlation between adverse versus control outcome groups, we observed a significant difference at the 10% and 20% thresholds, where a TC score above the threshold associated with a significantly less favorable outcome (p = 0.022 and p = 0.013, respectively). We observed no significant difference at the 30%, 40%, or 50% thresholds (p = 0.054, p = 0.15, and p = 0.38, respectively). The log-rank survival analysis showed a significant correlation between a reduction in RFS at the TC score thresholds of 10% (p = 0.044), 20% (p < 0.01), and 30% (p = 0.036; Fig 4). We observed no significant association with a shorter OS at any of the TC thresholds. When splitting the samples according to the TC score into three groups of <10%, 10–29%, and ≥30%, we found that a higher TC score significantly associated with a less favorable RFS (log rank = 0.038; Fig 5). In the Cox univariate regression analysis, the TC score correlated with a diminished RFS at thresholds 20% and 30% (HR = 2.46, p < 0.01 and HR = 1.84, p = 0.039, respectively) and for the ≥ 30% TC group in a three-group split using <10% TC as reference (HR = 7.48, p = 0.049). In Cox multivariate regression analysis adjusted for age, RFS was significantly reduced for a 20% threshold (HR = 2.47, p = 0.009), 30% threshold (HR = 1.83, p = 0.041) and for the ≥30% TC group in a three-group split using <10% TC as reference group (HR = 7.45, p = 0.049) (Table 3). A higher age at diagnosis did not significantly correlate with a higher TC score (Spearman's rho = -0.028, p = 0.80).

### 3.4. Visually assessed tall cell score and survival

When evaluated using 10% increments for the visual TC score, the median TC score for the controls was 0–9% and 30–39% for the adverse outcome group. Overall, the visual TC score was significantly higher in the adverse outcome group compared with the controls (p = 0.008). Examining the visual TC score in 10% increments, we observed a significant difference in the distribution between the adverse outcome group and controls when using the 10%, 20%, 30%, 40%, and 50% thresholds (p = 0.030, p = 0.031, p = 0.026, p = 0.030, and p = 0.024, respectively; Fig 4). We observed no significant correlation at the 60%, 70%, or 80% thresholds (p = 0.094,

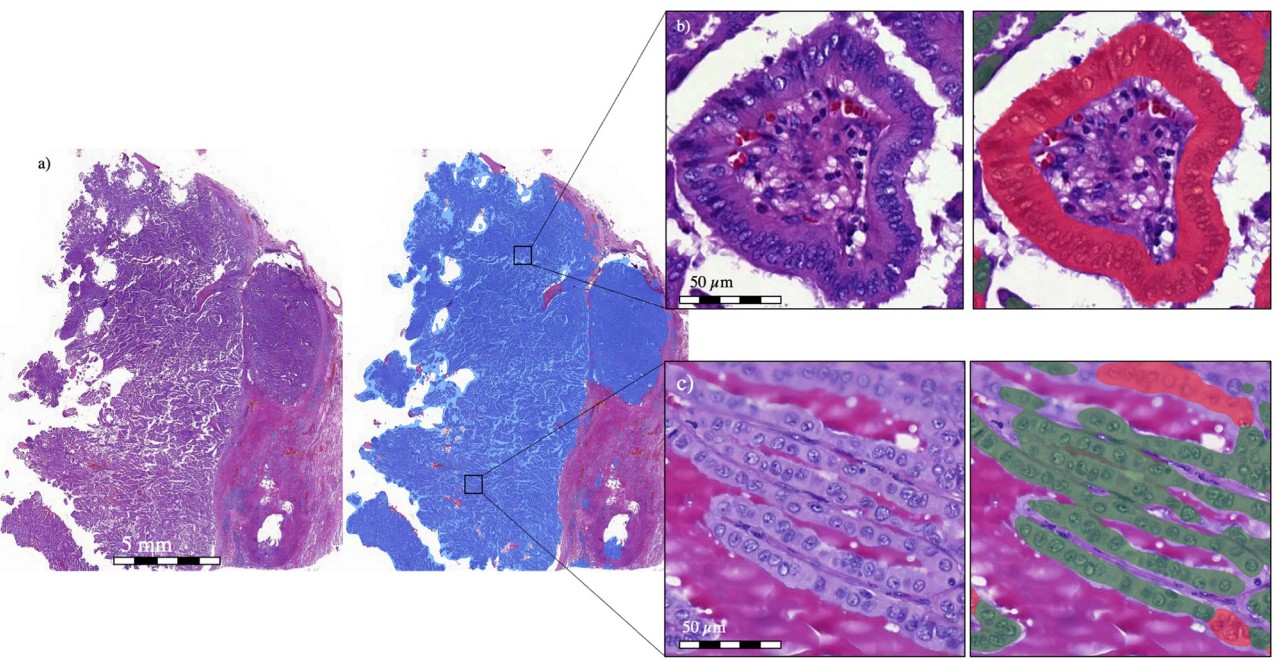

**Fig 3.** a) A zoomed-out view of a papillary thyroid carcinoma tissue sample in which the results of the deep learning (DL) algorithm's first layer are shown (blue). From the registered carcinoma area, the DL algorithm then registers the carcinoma epithelium as either b) tall cell (TC) (red) or c) non-TC area (green). The DL algorithm determines the percentage of the epithelium covered by TCs, that is, the TC score.

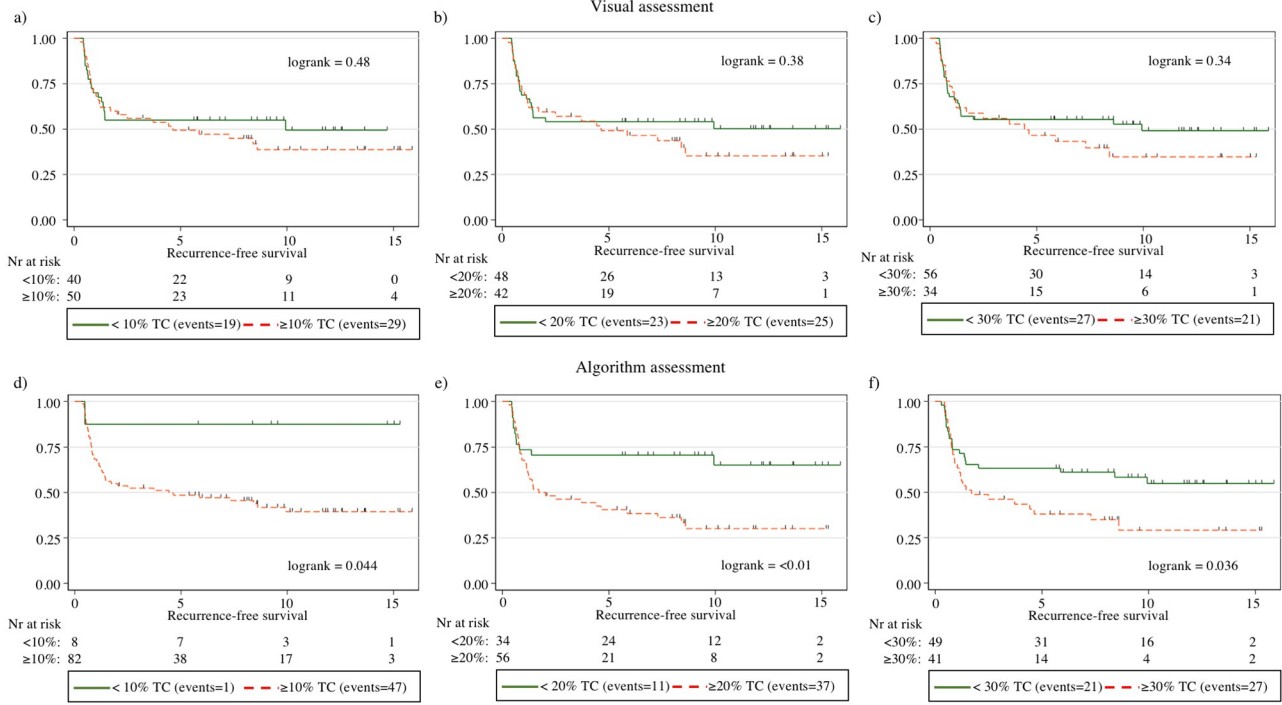

**Fig 4. Kaplan–Meier curves for relapse-free survival (RFS) among patients with papillary thyroid cancer according to three tall cell percentage thresholds: 10%, 20%, and 30% based on visual assessment (a–c) and using the algorithmic assessment (d–f).** In the figure, tall cell is abbreviated as TC.

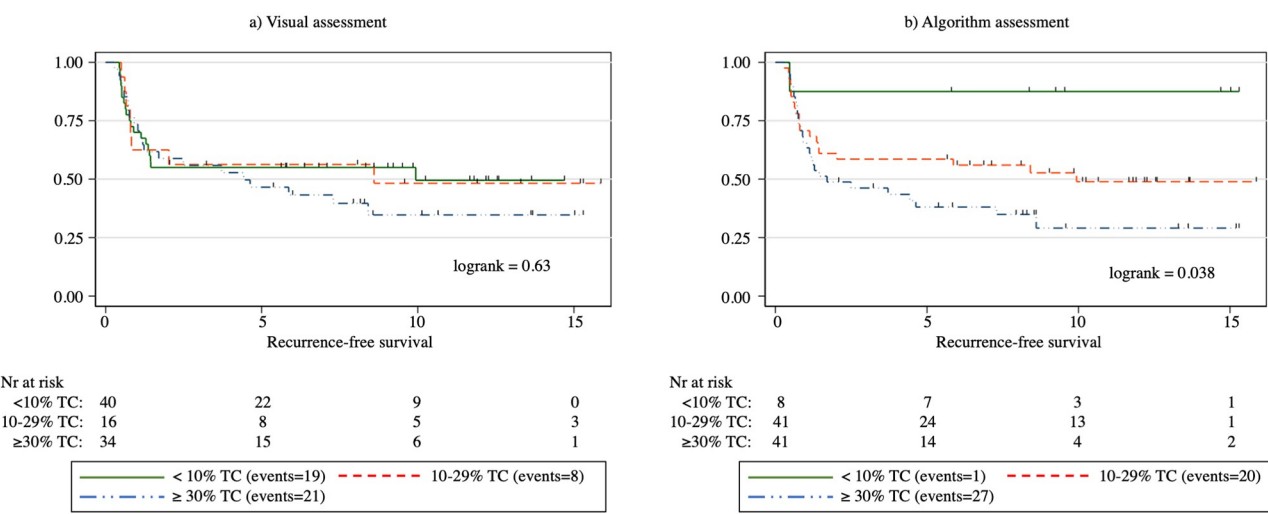

**Fig 5. Kaplan–Meier curves for relapse-free survival (RFS) among patients with papillary thyroid cancer according to tall cell percentage thresholds: <10%, 10–29%, and ≥30% based on a) visual assessment and b) algorithmic assessment.** In the figure, tall cell is abbreviated as TC.

**Table 3. Cox univariate and multivariate regression analysis for relapse-free survival (RFS) among patients with papillary thyroid carcinoma.**

| Parameters | Univariate analysis | | | Multivariate analysis | | |
|---|---|---|---|---|---|---|
| | Hazard ratio | p value | 95% CI | Hazard ratio | p value | 95% CI |
| Age >45 years | 0.91 | 0.74 | 0.51–1.62 | | | |
| Algorithm TC thresholds* | | | | | | |
| ≥10% | 6.00 | 0.076 | 0.83–43.6 | 5.93 | 0.079 | 0.81–43.2 |
| ≥20% | 2.46 | **0.009** | 1.25–4.86 | 2.38 | **0.013** | 1.20–4.70 |
| ≥30% | 1.84 | **0.039** | 1.03–3.27 | 1.83 | **0.041** | 1.02–3.26 |
| Algorithm TC three group split† | | | | | | |
| <10% | - | - | - | - | - | - |
| 10–29% | 4.80 | 0.13 | 0.64–35.7 | 4.81 | 0.13 | 0.64–35.8 |
| ≥30% | 7.48 | **0.049** | 1.01–55.2 | 7.45 | **0.049** | 1.01–55.0 |
| Multivariate adjusted for age‡ | | | | 0.96 | 0.88 | 0.53–1.71 |
| Visual TC thresholds* | | | | | | |
| ≥10% | 1.23 | 0.49 | 0.69–2.19 | 1.24 | 0.46 | 0.69–2.23 |
| ≥20% | 1.29 | 0.38 | 0.73–2.28 | 1.30 | 0.37 | 0.74–2.30 |
| ≥30% | 1.32 | 0.34 | 0.75–2.34 | 1.32 | 0.34 | 0.75–2.35 |
| Visual TC three group split† | | | | | | |
| <10% | - | - | - | - | - | - |
| 10–29% | 1.02 | 0.96 | 0.45–2.34 | 1.04 | 0.92 | 0.45–2.40 |
| ≥30% | 1.33 | 0.37 | 0.71–2.48 | 1.34 | 0.36 | 0.72–2.40 |
| Multivariate adjusted for age‡ | | | | 0.90 | 0.72 | 0.50–1.61 |

*The two group splits are analyzed for 10%, 20% and 30% TC thresholds and are studied as separate models as illustrated with dashed borders.

†<10% TC group used as reference group when analyzing the three-group split of <10% TC, 10–29% TC, and ≥30% TC.

‡Multivariate Cox regression analysis adjusted for age threshold of 45 years was analyzed according to the seventh edition of the American Joint Committee on Cancer staging of papillary thyroid cancer.

Abbreviation: TC, tall cell.

p = 0.19, and p = 0.14, respectively). The log-rank survival analysis revealed no significant correlation between the visually assessed TC score and RFS at any of the 10% increments in the TC score thresholds. Similarly, we observed no significant correlation between the visual TC score and overall survival. When split into three groups of <10%, 10–29%, and ≥30% TCs, we also detected no statistically significant association between the TC score and RFS (log rank = 0.63; Fig 5). In Cox univariate nor multivariate Cox regression analysis, none of the TC thresholds correlated with a reduction in RFS. Nor did the three-group split of <10% TC, 10–29% TC, and ≥30% TC correlate with a reduction in RFS (Table 3).

### 3.5. Tumor relapse samples

When visually evaluating the 71 slides from tumor relapses, the median TC score group was 10–19%. The median algorithm-based TC score was 27.3% (SD ± 11.45%), which is comparable to the 31.6% reported for primary tumors among the adverse outcome patients (p = 0.36). The agreement metric analysis yielded a weighted kappa value of 0.22 (SD ± 0.07) when patients were divided into TC score groups <10%, 10–29% and ≥30% (Table 4). When analyzing the TC score groups based on 10% increments, the agreement analysis yielded a weighted kappa value of 0.25 (SD ± 0.06; S2 Table in S1 File). The algorithm TC scoring of the relapse samples can be visually assessed via the following link: https://tinyurl.com/bde78hby.

## 4. Discussion

TCV PTC results in more adverse outcomes compared with the classical variant of PTC and should, therefore, be treated more aggressively [23]. Here, we present a TCV algorithm that quantifies the percentage of tumor epithelial area containing TCs in whole-slide images from H&E-stained PTC samples with a high sensitivity and specificity (https://tinyurl.com/bde78hby). Survival analysis demonstrated that the algorithm-based TC score significantly predicts relapse-free survival (RFS), whereas we detected no statistically significant association between the visually assessed TC score and RFS.

Inter- and intra-observer variability represents a major challenge in TCV classification [3, 14], and an improved reproducibility of TC identification and quantification is needed. Although the WHO defines TC as a cell that is two to three times as tall as it is wide, it is quite difficult for humans to strictly follow this rule when visually evaluating a PTC slide [13]. We hypothesized that using a DL algorithm to evaluate the TC score could meet the demand for a more objective and more consistent means of evaluating the presence and number of TCs in a sample. As debate continues related to the optimal cutoff point for visually assessed TCV [12], automated methods similar to that proposed here could be used to more systematically analyze and establish TC score cutoff points that provide clinically meaningful subgroups according to the proportion of TCs in PTC tissue samples.

The highest TC score given by the algorithm was only 52%, indicating a more conservative scoring compared to reliance on a human investigator. Interestingly, the TCV algorithm also

**Table 4. Interrater agreement between the visually assessed tall cell (TC) percentage and the algorithm-based TC score in tumor relapse samples.**

| | | Algorithm TC score assessment | | | Total |
|---|---|---|---|---|---|
| | | <10% | 10–29% | ≥30% | |
| Visual TC score assessment | <10% | 2 | 23 | 6 | 31 |
| | 10–29% | 0 | 7 | 9 | 16 |
| | ≥30% | 0 | 7 | 17 | 24 |
| | Total | 2 | 37 | 32 | 71 |

more rarely identified a tumor sample as having a low TC score of <10% (Table 2). Thus, it is possible that the TC score is both over- and underestimated through visual assessment. When studying TC thresholds using 10% increments, no statistically significant association was observed between the visually assessed TC score and recurrence-free survival at any of the thresholds. Using the algorithm-based TC score assessment, we observed a significant association between a higher TC score and a less favorable RFS using thresholds set to 10%, 20%, and 30% (Fig 4). This finding agrees with previous studies, emphasizing the clinical impact of a low percentage of TCs, while percentages as low as 10% should be reported by pathologists [5, 10]. In univariate Cox regression analysis, a significant reduction in RFS could be observed for both a 20% and 30% TC threshold for the algorithm TC scoring. In multivariate analysis we observed a significant reduction with a 20% and 30% threshold. In an age-adjusted multivariate Cox regression analysis of the three-group split of algorithm TC scoring using <10% TC as reference, we observed a significant reduction in RFS for the ≥30% TC group but not for the 10–29% TC (Table 3). These findings show that a 30% threshold indeed should be considered for diagnosing TCV, which is in line with WHO's recommendations [9]. The findings also suggests that cases with 10–29% TCs (HR = 4.81) could have a worse prognosis compared to <10% TC but the sample size of the current study could not prove this in a multivariate Cox regression analysis (p = 0.13) (Table 3).

Patients in the validation case–control cohort were all diagnosed within the same hospital district. Thus, they were all offered similar initial treatment—that is, surgery in combination with radioiodine ablation therapy. This rather aggressive initial treatment protocol could explain why we found so few cases of PTC with an adverse outcome when collecting data retrospectively. Aggressive disease was classified as a tumor relapsing at least twice. Such cases could potentially have been missed or even misclassified as control cases due to the initial aggressive treatment protocol. This could be considered a limitation of the present study, and, in future studies, we recommend carrying out a multicenter study to limit the impact of treatment protocols adopted by specific hospital districts.

In the present study, we used a commercially available image management and machine learning platform (Aiforia Create, Aiforia Technologies Oy, Helsinki, Finland), The exact architecture of the algorithm is proprietary and thus could not be reported which is a limitation. However, the same platform can be used in future studies to fully reproduce the experiment.

Previously, little research focused on the morphologies of PTC metastases. Therefore, we also assessed the TC score in 71 metastatic samples obtained from 32 patients with relapses, that is, aggressive disease, using both visual evaluation and the algorithm-based TC score. We hypothesized that a higher TC score could be seen in samples taken from metastatic tissue. This, however, seems not to be the case, since we observed no statistically significant difference between the median TC score in the primary tumor versus the relapse tumors (27.3% and 31.6%, respectively, p = 0.36). This result suggests that the TCV morphology is retained in metastatic tissue as well. Thus, TC score quantification could possibly also be completed for metastatic samples.

One strength to this study lies in the carefully matched adverse outcome case–control cohort we used as the validation dataset. Patients were matched by age of diagnosis (within 10 years), sex, and tumor stage. All patients in the validation dataset were treated within the same hospital district and offered a similar initial treatment. However, adhering to such stringent criteria also limited the cohort size to only 90 patients. Since death from disease is a rare outcome in PTC, we defined an adverse outcome as having at least two relapses, distant metastases at primary diagnosis, or death from disease. The eligibility criteria of including patients with two relapses in the dataset representing an adverse PTC could mean that we might have allowed more benign cases to be included as aggressive cases. To only use death from disease

as the single criteria for aggressive disease is preferable but leads to a low number of events in single-center series. Another strength lies in the training dataset used for the TC score algorithm. During training, we used a previous PTC series of 70 WSIs. Furthermore, we included 30 WSIs from the TCGA database in order to broaden the spectrum of stain variations in the training set and, thus, improve the generalizability of the TCV algorithm. The colors and contrasts in the selected H&E stained TCGA WSIs visually differed from the corresponding H&E-stained slides in the Helsinki series used to train the algorithm. Furthermore, these WSIs contained morphologies of non-TCV PTC as well as those of TCV. To address the potential variability in the sample properties, we utilized both the color and scale augmentations of the training samples to improve the generalizability. This resulted in a high sensitivity and specificity in the independent test set and also visually accurate segmentation of both TC and non-TC tumor epithelium in the validation set. However, we expect that the performance will decline when testing algorithms on samples from different centers [24]. Thus, in future studies, the performance of the trained TC score algorithm requires validation on external datasets within a multicenter validation study.

In conclusion, we show that the DL-based algorithm was better than the human observer in identifying TCVs. The algorithm could prove useful as a clinical tool for pathologists when evaluating PTC samples and can potentially significantly improve the consistency of TCV case assessment. To our knowledge, no such algorithm has previously been described. The results indicate that a 30% threshold should be used in diagnosing TVC. However, all cases with more than 10% TCs should be included in the pathologist's reports. Our results also suggest that a higher TC score in PTC assessed using the DL algorithm is associated with less favorable survival. Finally, we show that TC morphology is retained, although the proportion of the TC area does not increase in the tumor tissue from relapsed patients, suggesting that the diagnosis of TCV can rely on metastatic tissue as well.

## Supporting information

**S1 File.**
(DOCX)

## Acknowledgments

We thank the FIMM Digital Microscopy and Molecular Pathology Unit supported by the Helsinki Institute of Life Science and Biocenter Finland for excellent assistance. The results reported here are in part based upon data generated by the TCGA Research Network (https://www.cancer.gov/tcga).

## Author Contributions

**Conceptualization:** Sebastian Stenman, Caj Haglund, Johanna Arola, Johan Lundin.

**Data curation:** Sebastian Stenman, Mikael Lundin.

**Formal analysis:** Mikael Lundin, Johanna Arola.

**Funding acquisition:** Caj Haglund, Johanna Arola, Johan Lundin.

**Investigation:** Sebastian Stenman, Johanna Arola.

**Methodology:** Sebastian Stenman, Nina Linder.

**Project administration:** Caj Haglund, Johan Lundin.

**Resources:** Johan Lundin.

**Software:** Mikael Lundin, Johan Lundin.

**Supervision:** Nina Linder, Caj Haglund, Johanna Arola, Johan Lundin.

**Writing – original draft:** Sebastian Stenman.

**Writing – review & editing:** Nina Linder, Caj Haglund, Johanna Arola, Johan Lundin.

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
