## [Decision Letter · Decision Letter 0]

24 Aug 2021

PONE-D-21-21384

A deep learning–based algorithm for tall cell detection in papillary thyroid carcinoma

PLOS ONE

Dear Dr. Stenman,

Thank you for submitting your manuscript to PLOS ONE. After careful consideration, we feel that it has merit but does not fully meet PLOS ONE’s publication criteria as it currently stands. Therefore, we invite you to submit a revised version of the manuscript that addresses the points raised during the review process.

ACADEMIC EDITOR: There are still some minor issues requiring to be addressed. Please kindly respond to the reviewers' comments. 

We look forward to receiving your revised manuscript.

Kind regards,

Jason Chia-Hsun Hsieh, M.D. Ph.D

Academic Editor

PLOS ONE

4. Please include a copy of Table 2 which you refer to in your text on page 8.

5. We note you have included a table to which you do not refer in the text of your manuscript. Please ensure that you refer to Table 5 in your text; if accepted, production will need this reference to link the reader to the Table.

Additional Editor Comments (if provided):

There are still some minor issues requiring to be addressed. Please kindly respond to the reviewers' comments.

Reviewers' comments:

Reviewer's Responses to Questions

**Comments to the Author**

1. Is the manuscript technically sound, and do the data support the conclusions?

Reviewer #1: Yes

Reviewer #2: No

2. Has the statistical analysis been performed appropriately and rigorously? 

Reviewer #1: Yes

Reviewer #2: No

3. Have the authors made all data underlying the findings in their manuscript fully available?

Reviewer #1: No

Reviewer #2: Yes

4. Is the manuscript presented in an intelligible fashion and written in standard English?

Reviewer #1: Yes

Reviewer #2: Yes

5. Review Comments to the Author

Reviewer #1: General Comments: The authors aimed to train a deep learning algorithm to detect the proportion of tall cell variants in papillary thyroid carcinomas. Comparisons were made to scoring by a human investigator, 71 samples were used for training and testing, with an additional 90 samples used for validation.

Specific Comments:

1. Methods, Patient Series, Training Series: The algorithm is reported to have been trained on 100 whole slide images (70 from HUH; 30 from TCGA), however it is not explicitly stated whether these are from separate individuals. Please indicate if these are from separate subjects and, if not, how many unique individuals provided samples.

2. Methods, Patient Series, Testing Series: The authors include a separate validation set of 90 patients and were careful to include both cases (with adverse outcomes) and controls. This is sound practice. However, the authors mention that 71 tissue samples were obtained only from those in the adverse outcome group. It is not clear whether the validation sample size is 90 or 71, and if the latter, what the point of including the controls was.

3. Methods, Training of Deep Learning Algorithm: The authors mention 2,674 manual annotations were used for training and 296 were used for testing. The large numbers involved help justify the somewhat extreme 90/10 training/testing set ratio (something like 70/30 is more standard), but it is not clear where these large numbers come from, given that the previously reported sample size were 90.

4. Methods, Training of Deep Learning Algorithm: The authors performed their training/testing split once and missed an opportunity to determine the stability of their algorithm and the validity of their results by performing different splits of the data into training/testing and reproducing the model. Please justify why only a single split was made.

5. Methods, Statistical Analysis: The methods appear sound, though they are slightly under-reported. For instance, the log-rank test and proportional hazard regression model are used for survival analysis, but for what purpose? Please specify the outcomes and fixed effects, as well as the goals for these analyses. The reporting for exact tests and kappa statistics in the same paragraph are excellent examples of the detail needed.

6. Methods, Statistical Analysis: There is shockingly little detail provided about the neural network / deep learning approach. Input and output nodes, the number of hidden layers, the number of neurons, are not reported or discussed. Please provide these details – needed for reproducibility – along with justification for those choices. A Figure would also be helpful.

7. Results: In general, the findings are thoroughly and adequately reported. It is not precisely clear what role the validation sample played in generating them. Please state which results are from testing and which are from validation.

8. Results: The authors do not appear to have calculated sensitivity and specificity for the validation set. This seems an important omission. Please report or state how and why the data preclude such calculations.

9. Discussion: The authors state several strengths of their work without explicitly stating any limitations. Surely there must be some!

Reviewer #2: Interesting premise, but I have the following questions about study design:

In the validation set all samples were from patients with at least 5 years follow up. This means that any patients with a cancer with a death or other event that lead to loss of follow up by 5 years were not included, thus the population was a healthier population that the expected population with this cancer. How was this bias addressed in the study?

Patients with an adverse outcome - what was the time frame for this observation?

How was the matching accounted for in the analyses?

How was median relapse free survival assessed? What events were included/ censoring events?

Table 1: include percentages to show if the rates were similar in the two groups for categorical variables.

Definition of overall survival?

6. PLOS authors have the option to publish the peer review history of their article (what does this mean?). If published, this will include your full peer review and any attached files.

Reviewer #1: No

Reviewer #2: No

---

## [Author Response · Author response to Decision Letter 0]

10 Nov 2021

Rebuttal letter 

Please find the comments of the reviewers and the rebuttal below:

Reviewer #1: 

General Comments: The authors aimed to train a deep learning algorithm to detect the proportion of tall cell variants in papillary thyroid carcinomas. Comparisons were made to scoring by a human investigator, 71 samples were used for training and testing, with an additional 90 samples used for validation.

The primary aim of the study was to train a deep learning-based algorithm to detect tall cells in papillary thyroid carcinoma (PTC) and compare the algorithm results to that of a human investigator as well as the outcome of the patients. A total of 100 whole-slide images (WSI) from 100 separate PTC patients were used for training and testing the algorithm; 70 patients and corresponding WSIs from the Hospital District of Helsinki and Uusimaa and the remaining 30 patients and WSIs from the Cancer Genome Atlas Program (TCGA). The validation series consisting of 90 PTC WSIs from 90 separate age-, sex, and T-stage matched PTC patients was used to validate the trained algorithm and how it performed compared to a human investigator. 

Furthermore 71 tissue slides from 71 separate relapses from the 34 patients with an aggressive disease (figure 1) in the validation series were analysed. As a secondary aim of the study, these tissue slides representing cases with relapse were also assessed by the algorithm as well as the human investigator to study whether the TC morphology of the primary tumor could also be seen in the relapse morphology. 

Comment: 1. Methods, Patient Series, Training Series: The algorithm is reported to have been trained on 100 whole slide images (70 from HUH; 30 from TCGA), however it is not explicitly stated whether these are from separate individuals. Please indicate if these are from separate subjects and, if not, how many unique individuals provided samples.

Rebuttal: The 100 total samples in the training set all originated from separate patients. 

All formalin-fixed paraffin embedded (FFPA) tissue samples from all 70 patients originating from the Hospital District of Helsinki and Uusimaa (HUS) were collected and reviewed by two researchers (S.S., J.A.). One representative FFPA tissue block was selected for each of the patients. New tissue slides were cut, stained with hematoxylin and eosin (HE) and further digitally scanned and imported to the image management platform used to train the algorithm (Aiforia, Aiforia Technologies Oy, Helsinki, Finland). In order to improve the generalizability of the trained algorithm, 30 additional tissue slides originating from the Cancer Genome Atlas Program (TCGA). These 30 slides were from cases with separate patients and were chosen based on visual assessment. Please see the CONSORT diagram of the different datasets used in the study (figure 1). 

Adjustments made: Manuscript edited, and figure added to clarify that the training samples originated from separate patients (page 3, Methods section, 2.1.1. Training series, row 2 and the newly added figure (figure 1) CONSORT flow diagram on page 5, Methods section, 2.3. Training of the deep learning algorithm, rows 3-5, figure 1, page 4). 

Comment: 2. Methods, Patient Series, Testing Series: The authors include a separate validation set of 90 patients and were careful to include both cases (with adverse outcomes) and controls. This is sound practice. However, the authors mention that 71 tissue samples were obtained only from those in the adverse outcome group. It is not clear whether the validation sample size is 90 or 71, and if the latter, what the point of including the controls was.

Rebuttal: The validation set included 90 patients; 34 patients with an adverse outcome and 56 age-, gender-, and T stage matched control cases.

The main objective of the study was to validate the trained algorithm using a case-control approach and to study how the TC score of the algorithm correlated with disease outcome. A secondary objective of the study was to test a hypothesis that the tall cell composition is increased in relapse samples compared to the corresponding primary tumor sample. In order for us to analyse this, we collected all relapse/metastasis FFPA tumor samples (e.g. local tumor recurrence or lymph node metastases) from the 34 patients with an adverse outcome. In the same manner as with the primary tumor samples, all FFPA samples were reviewed by two researchers (S.S., J.A.) and one representative FFPA tissue block was selected for each of the 71 relapses. The FFPA were then cut and the tissue samples were stained with HE digitized and imported to an image management platform. After the algorithm was validated on the 90 primary tumor samples in the validation set, the algorithm then analysed the 71 relapse tumor samples. In the study, we concluded that the tall cell morphology is retained but not increased in relapse samples compared to the primary tumor morphology. 

Adjustments made: Text edited to outline the aims of the study more clearly (Page 3, Introduction section, rows 22-24). A new subheader “Relapse series” was inserted to highlight that these WSIs are indeed not included in the validation series (page 5, Methods section, 2.1.3. Relapse series). Also, the “digitization of slides” subheader was deleted and embedded into the “Validation series” text (Page 4, Methods section, 2.1.2 Validation series, rows 7-16). A consort diagram added to illustrate the different datasets more clearly (figure 1).

Comment: 3. Methods, Training of Deep Learning Algorithm: The authors mention 2,674 manual annotations were used for training and 296 were used for testing. The large numbers involved help justify the somewhat extreme 90/10 training/testing set ratio (something like 70/30 is more standard), but it is not clear where these large numbers come from, given that the previously reported sample size were 90.

Rebuttal: In total, the training set consisted of 100 tumor samples; 70 samples originating from Helsinki University Hospital and the remaining 30 from the TCGA database. However, the algorithm was trained on 2,674 manual annotations of regions of interest within these 100 patient samples. The manual annotations were done by one researcher (S.S.). 

The reviewer brings up a good point that a training/testing set ratio or 80/20 or even 70/30 is more common and perhaps more standard. However, in a 90/10 split was justified based on the quantity of manual annotations in the training set, were as many as 2,674 and that the test set consisted of 296 manual annotations of regions of interest. A power calculation is reported below to illustrate that for analysis of the sensitivity of the algorithm on the annotation level, this number can be considered sufficient.

Changes made: Added a new figure of a CONSORT flow diagram outlining the datasets used in the study (figure 1, page 4). 

Comment: 4. Methods, Training of Deep Learning Algorithm: The authors performed their training/testing split once and missed an opportunity to determine the stability of their algorithm and the validity of their results by performing different splits of the data into training/testing and reproducing the model. Please justify why only a single split was made.

Rebuttal: If we assume the prevalence of TCs to be 30% (estimated based on visual assessment) in the manual annotations used for training and testing, we would need 244 manual annotations in the test set to reach the acceptable +/-5% width of the 95% confidence level of the estimated sensitivity if the expected sensitivity is 95% according to our calculations. Thus, the 296 manual annotations exceed this number and we therefore deemed that one split sufficed. Given the sensitivity level for detection of TC in the test set, we expect a similar sensitivity level i.e. 95% in the validation set. However, the reviewer is correct in that multiple splits could perhaps have determined the stability of the model in more detail. 

Comment: 5. Methods, Statistical Analysis: The methods appear sound, though they are slightly under-reported. For instance, the log-rank test and proportional hazard regression model are used for survival analysis, but for what purpose? Please specify the outcomes and fixed effects, as well as the goals for these analyses. The reporting for exact tests and kappa statistics in the same paragraph are excellent examples of the detail needed.

Rebuttal: As death from disease is a rather rare event in PTC, relapse was used as the primary outcome in survival analysis of the data. An event in relapse-free survival was defined as time from diagnosis to relapse (Increase in serum thyroglobulin levels or histological confirmation). Patients who died of other causes or otherwise were censored (e.g. treatment moved to another hospital district). 

Changes made: Clarification of parameters used in survival analysis added (page 7, Methods section, 2.4. Statistical analysis, rows 9-12)

Comment: 6. Methods, Statistical Analysis: There is shockingly little detail provided about the neural network / deep learning approach. Input and output nodes, the number of hidden layers, the number of neurons, are not reported or discussed. Please provide these details – needed for reproducibility – along with justification for those choices. A Figure would also be helpful.

Rebuttal: The exact architecture of the artificial neural network is not available and remains proprietary to the provider of the software (Aiforia Technologies Oy, Helsinki, Finland). This could be considered a limitation and has been added as such in the manuscript. However, as the platform is commercially available, it is possible for other researchers to access and use the platform for training of deep learning-based algorithms. Also, the image augmentation details are provided in the manuscript which makes the method fully reproducible. 

Comment: 7. Results: In general, the findings are thoroughly and adequately reported. It is not precisely clear what role the validation sample played in generating them. Please state which results are from testing and which are from validation.

Rebuttal: The validation set in the study was used for survival statistics and to compare how the algorithm’s scores compared to that of a human investigator. The validation set was a held-out set and thus not included in training or testing of the algorithm. 

Changes made: Added figure 1, page 4, and edited the text according to earlier comments to clarify this. 

Comment: 8. Results: The authors do not appear to have calculated sensitivity and specificity for the validation set. This seems an important omission. Please report or state how and why the data preclude such calculations.

Rebuttal: The sensitivity and specificity was calculated based on the manual annotations in the test set. The validation set had no ground truth TC labels, and a sensitivity and specificity could therefore not be calculated in the validation dataset. 

Currently, the TC score is assessed visually by the pathologists mostly using traditional microscopes. The assessment is done visually on slide level, as it would be near impossible for a human to assess the TC score on cell level throughout the whole slide. The TC score is then usually reported with 10% increments. The human TC scoring in the study was performed in the same way; the representative slides were evaluated and a TC score was reported for each representative slide. The algorithm in the study consisted of two layers; first, tumor tissue was detected and secondly both TC area and non-TC area were registered within the tumor tissue area. The second layer of the algorithm analyzed the tissue on a much higher magnification resulting in cell level analysis. That is why, in the validation set, no ground truth annotations existed and therefore no specificity nor sensitivity could be calculated. 

Comment: 9. Discussion: The authors state several strengths of their work without explicitly stating any limitations. Surely there must be some!

Rebuttal: The validation series of the study were all collected from within the same hospital district. During the time of sample collection, the hospital district offered a rather aggressive initial treatment protocol to adequately treat all aggressive cases but maybe overtreated the more indolent PTCs. This, in turn, could lead to misclassifying of the cases included in the study. As mentioned in the discussion section, we highly recommend multicenter studies in the future to limit the impact of treatment protocols within a specific hospital districts. 

Adjustments made: Explicitly mentioning the limitation of the study cohort (Page 12, discussion section, rows 7-8)

 

Reviewer #2: 

Interesting premise, but I have the following questions about study design:

Comment: In the validation set all samples were from patients with at least 5 years follow up. This means that any patients with a cancer with a death or other event that lead to loss of follow up by 5 years were not included, thus the population was a healthier population that the expected population with this cancer. How was this bias addressed in the study?

Rebuttal: The patients in the validation set included patients that were diagnosed with papillary thyroid carcinoma (PTC) between 2003-2013. By design, the follow-up time extended to 2018, which allowed for all patients included in the study to have a 5-year follow-up time. Patients who died of other causes than PTC or for other causes were censored within 5 years from diagnosis date were also included in the study. This is how, even though we allowed for a minimum 5-year follow-up, some patients had a shorter follow-up and the range of follow-up for the study was 2.1 years to 15.8 years (median 10.1 years) which is stated in the manuscript (page 5, Methods, Validation series, rows 6-7). 

Adjustments made: Text edited to clarify the follow-up time (page 4, Methods section, 2.1.2. Validation series, rows 2-3)

Comment: Patients with an adverse outcome - what was the time frame for this observation?

Rebuttal: In the present study, an adverse outcome was defined as PTC cases having at least two recurrences (histological confirmation or serum thyroglobulin elevation during follow-up), having distant metastases or patients who died from PTC. Distant metastases at primary diagnosis was classified as an adverse outcome disease. A PTC was also defined as having an adverse outcome if the PTC distally metastasized, recurred more than twice or the patient died of the disease during follow-up. 

Comment: How was the matching accounted for in the analyses?

Rebuttal: Both the human investigator and the deep learning-based algorithm were blinded to the clinical outcome and thus also to the matching when giving TC scores to the cases. 

Comment: How was median relapse free survival assessed? What events were included/ censoring events?

Rebuttal: Relapse free survival was defined as time from diagnosis to first relapse. A relapse was defined as serum thyroglobulin elevation during follow-up or histological confirmation. Patients for whom follow-up ended before relapse for different reasons (e.g. moving to another hospital district or death from other disease) were censored in the survival analysis.

Adjustments made: Clarification of text on page 7, Methods section, 2.4. Statistical analysis, rows 9-12)

Comment: Table 1: include percentages to show if the rates were similar in the two groups for categorical variables.

Changes made: Percentages added to appropriate variables in table 1. 

Comment: Definition of overall survival?

Rebuttal: Overall survival was defined as time from diagnosis to death from any cause. 

Adjustments made: Clarification of text on page 7, Methods section, 2.4. Statistical analysis, rows 9-12)

---

## [Decision Letter · Decision Letter 1]

17 Jan 2022

PONE-D-21-21384R1A deep learning–based algorithm for tall cell detection in papillary thyroid carcinomaPLOS ONE

Dear Dr. Stenman,

Thank you for submitting your manuscript to PLOS ONE. After careful consideration, we feel that it has merit but does not fully meet PLOS ONE’s publication criteria as it currently stands. Therefore, we invite you to submit a revised version of the manuscript that addresses the points raised during the review process.

ACADEMIC EDITOR: We are sorry for the delayed evaluation. Some questions require further clarification. 

We look forward to receiving your revised manuscript.

Kind regards,

Jason Chia-Hsun Hsieh, M.D. Ph.D

Academic Editor

PLOS ONE

Additional Editor Comments (if provided):

Some questions require further clarification.

Reviewers' comments:

Reviewer's Responses to Questions

**Comments to the Author**

1. If the authors have adequately addressed your comments raised in a previous round of review and you feel that this manuscript is now acceptable for publication, you may indicate that here to bypass the “Comments to the Author” section, enter your conflict of interest statement in the “Confidential to Editor” section, and submit your "Accept" recommendation.

Reviewer #1: (No Response)

Reviewer #2: (No Response)

2. Is the manuscript technically sound, and do the data support the conclusions?

Reviewer #1: Yes

Reviewer #2: Partly

3. Has the statistical analysis been performed appropriately and rigorously? 

Reviewer #1: Yes

Reviewer #2: No

4. Have the authors made all data underlying the findings in their manuscript fully available?

Reviewer #1: Yes

Reviewer #2: Yes

5. Is the manuscript presented in an intelligible fashion and written in standard English?

Reviewer #1: Yes

Reviewer #2: Yes

6. Review Comments to the Author

Reviewer #1: While the authors responded to my comment (#6, reproduced below), they did not in fact mention the proprietary nature of the neural network approach in the limitations section.

Comment: 6. Methods, Statistical Analysis: There is shockingly little detail provided

about the neural network / deep learning approach. Input and output nodes, the

number of hidden layers, the number of neurons, are not reported or discussed. Please

provide these details – needed for reproducibility – along with justification for those

choices. A Figure would also be helpful.

Rebuttal: The exact architecture of the artificial neural network is not available and

remains proprietary to the provider of the software (Aiforia Technologies Oy, Helsinki,

Finland). This could be considered a limitation and has been added as such in the

manuscript. However, as the platform is commercially available, it is possible for other

researchers to access and use the platform for training of deep learning-based

algorithms. Also, the image augmentation details are provided in the manuscript which

Powered by Editorial Manager® and ProduXion Manager® from Aries Systems Corporation

makes the method fully reproducible.

Reviewer #2: Regarding the comment of:

"How was the matching accounted for in the analyses?" - this requires a statistical analyses response. For example - In the Cox proportional hazards regression, adjustment was made for the matching variables of age, gender, tumour stage...

Table 3 could be updated to include multiple variable analyses adjusted for age group, gender, tumour stage.

Is it possible that some controls would be cases if observed for a longer period of time? How did 14 patients experience relapse, but were considered controls (if followed for longer, could a second relapse have been observed, making them cases)? It seems that the two groups were linked with the relapse free survival time - thus any analysis of the effect of group on relapse free survival would be biased. It would not be possible to define which group a patient would fall into at baseline.

In table 1 median age at diagnosis for cases is 43.4 years, but in text above it says 41.0 years - which is correct? Similar 41.7 and 41.5 are both quoted for the controls - which is right?

Figure 5: the third line in the number at risk - the label should be >=30% rather than 10% to match with the legend.

Include in table 3 the three group split as shown in figure 5, including univariate and multiple variable analyses with this grouping.

The question is - is a three group split better than a two group split - and this is difficult to tell with these analyses.

In the definition of relapse free survival I assume that relapse and death are events, end of follow up is a censoring variable? It is not clear in this sentence.

7. PLOS authors have the option to publish the peer review history of their article (what does this mean?). If published, this will include your full peer review and any attached files.

Reviewer #1: No

Reviewer #2: No

---

## [Author Response · Author response to Decision Letter 1]

3 Mar 2022

Rebuttal letter

Reviewer #1:

While the authors responded to my comment (#6, reproduced below), they did not in fact mention the proprietary nature of the neural network approach in the limitations section.

Rebuttal: The architecture of the algorithm is not available in exact detail, so it would indeed not be possible to recreate the model without knowing the algorithm structure. With that said, the image management and machine learning platform used in the study is commercially available for anyone to use. As the parameter settings that can be changed in the platform (e.g., feature size, iterations, image augmentations etc.) have been described in detail in the manuscript, the method is considered fully reproducible. 

Changes made: It is now mentioned that the exact architecture of the algorithm is proprietary and that it could be considered a limitation. (Page 12, Discussion section, rows 26-29) 

Reviewer #2: 

1. "How was the matching accounted for in the analyses?" - this requires a statistical analyses response. For example - In the Cox proportional hazards regression, adjustment was made for the matching variables of age, gender, tumour stage...

Table 3 could be updated to include multiple variable analyses adjusted for age group, gender, tumour stage.

Rebuttal: The matching of the cases was done when collecting the samples for the study. Thus, the entire study, including the statistical analyses, were affected and accounted for in the study. In Cox multivariate regression analysis, the model could not be fitted with all four variables the reviewer suggested due to diminishing power and overfitting. We performed a Cox multivariate regression analysis adjusted for age (at 45 years of age threshold) and updated the manuscript and table 3 accordingly. 

Changes made: Included the recent changes in the manuscript and updated table 3 according to the findings. (Page 9, results section, 3.3. Algorithm-based tall cell score and survival subsection, rows 8-11 and 3.4 Visually assessed tall cell score and survival, rows 10-11, page 12, Discussion section, rows 11-18. Table 3, page 10, reworked and corrected)

2. Is it possible that some controls would be cases if observed for a longer period of time? How did 14 patients experience relapse, but were considered controls (if followed for longer, could a second relapse have been observed, making them cases)? It seems that the two groups were linked with the relapse free survival time - thus any analysis of the effect of group on relapse free survival would be biased. It would not be possible to define which group a patient would fall into at baseline.

Rebuttal: Papillary thyroid carcinoma (PTC) often has a great prognosis, even with local recurrence. Thus, we designed the present study such that an adverse outcome was defined as two or more relapses (thyroglobulin elevation or histological confirmation), distant metastasis (at primary diagnosis or during follow-up) or death from PTC. Each case with an adverse outcome was matched with one to two control cases matched by age at diagnosis (within 10 years), gender, and tumor stage. The cases were collected between 2003 and 2013 and follow-up was ended 2018. This allowed all included cases at least a 5-year follow-up. By design, this cohort includes a concentrated number of cases with an aggressive disease and as such does not represent the general population. As tall cell variants are rather rare, the concentration of aggressive cases allowed us to study the prognostic impact of tall cells with a rather small sample size which was the aim of the study design. 

In the survival analysis the cohort was analyzed as a whole and not grouped by aggressive or control. For example, analyzing a 30% tall cell threshold, the only grouping was whether the patient had less or more than 30% tall cells. Thus, the aggressive vs. control grouping would not affect the survival analysis of tall cell thresholds. 

We hypothesized that patients with a single local recurrence does not have an aggressive disease and that a single recurrence could also be a result of inadequate resection of tumor in the primary surgery. It is possible that some of the patients in the control group could have been diagnosed with additional recurrences if observed for a longer period and longer follow-up periods are thus generally recommended. However, we felt that a 5-year follow-up was adequate for the scope of this study. 

The cohort was retrospectively collected and most of the criteria for an aggressive disease indeed occurred during follow-up and would have been unknown at time of diagnosis. The reviewer correctly points out that it would not have been possible to define which group a patient would fall into at the time of diagnosis based on the aggressive disease criteria used in the study. However, our aim was to examine tall cell thresholds and how the tall cell scores correlated with outcome. As we concluded in the study, a 30% tall cell threshold did indeed correlate with a reduced relapse free survival. This could have been observed at the time of diagnosis as we studied the same histological tissue blocks that were used in the initial diagnosis. We argue that cases with 30% TC or higher should be treated more aggressively as they correlate with a more aggressive disease. 

In the future, the algorithm presented here could be used by pathologists in the diagnosis of tall cell variants of papillary thyroid carcinoma and further studied using a prospective cohort. 

Changes made: Discussion and conclusion sections edited to highlight the findings in the survival analysis (page 12, Discussion section, rows 11-18 and page 13-14, Discussion section, last paragraph of the manuscript)

3. In table 1 median age at diagnosis for cases is 43.4 years, but in text above it says 41.0 years - which is correct? Similar 41.7 and 41.5 are both quoted for the controls - which is right?

Rebuttal: Both numbers are correct since different numbers are reported; In the text, the median age is reported, while in the table the mean age is reported. This understandably cause confusion and we will edit the manuscript to be more consistent. 

Changes made: corrected the table so the median age is reported instead of the mean to avoid confusion (page 5, table 1, row 11)

4. Figure 5: the third line in the number at risk - the label should be >=30% rather than 10% to match with the legend.

Rebuttal: The reviewer correctly points out an error in the figure.

Changes made: Figure 5 corrected to match the figure legend. 

5. Include in table 3 the three group split as shown in figure 5, including univariate and multiple variable analyses with this grouping.

Changes made: Table 3 changed according to the reviewer’s suggestion. (page 10)

6. The question is - is a three group split better than a two group split - and this is difficult to tell with these analyses.

Rebuttal: The WHO suggests using a 30% TC threshold, thus basically recommending using a two-group split in clinical practice. This is probably preferable in clinical practice as it is very difficult for human observers to adhere to the strict three-group split used in the study (under 10%, 10-29%, and over 30%).

The idea of the three-group split in the statistical analysis was to separate the cases with a very small number of TCs (under 10%) which we assumed would have the best prognosis. This seems to be a valid assumption based on figure 5. However, only 8 cases had less than 10% TCs and only one relapse was observed within this group. Having separated out the cases with very few TCs, we could then study the difference between some TCs (10-29%) versus the cases with a lot of TCs (over 30%). Again, based on figure 5, there seem to be a difference in relapse between these two groups as well. 

The reviewer asks a crucial question and to summarize; Based on our results, when using the algorithm, a 30% TC threshold seem to be the correct threshold to use when diagnosing TCV, which is in line with WHOs current recommendations. However, all cases with TCs (≥10%) should be included in the pathologist’s report as these cases also seem to correlate with a reduction in relapse free survival.

Changes made: Conclusion edited to highlight these findings (Page 13-14, Discussion section, last paragraph of the manuscript).

7. In the definition of relapse free survival I assume that relapse and death are events, end of follow up is a censoring variable? It is not clear in this sentence.

Rebuttal: relapse free survival is the time from diagnosis to relapse as described on page 7, Methods section, 2.4 statistical analysis subsection rows 9-10. The event is thus relapse and censoring events are e.g., end of follow-up or death.

---

## [Decision Letter · Decision Letter 2]

8 Jun 2022

PONE-D-21-21384R2A deep learning–based algorithm for tall cell detection in papillary thyroid carcinomaPLOS ONE

Dear Dr. Stenman,

Thank you for submitting your manuscript to PLOS ONE. After careful consideration, we feel that it has merit but does not fully meet PLOS ONE’s publication criteria as it currently stands. Therefore, we invite you to submit a revised version of the manuscript that addresses the points raised during the review process.

We look forward to receiving your revised manuscript.

Kind regards,

Alvaro Galli

Academic Editor

PLOS ONE

Journal Requirements:

Reviewers' comments:

Reviewer's Responses to Questions

**Comments to the Author**

1. If the authors have adequately addressed your comments raised in a previous round of review and you feel that this manuscript is now acceptable for publication, you may indicate that here to bypass the “Comments to the Author” section, enter your conflict of interest statement in the “Confidential to Editor” section, and submit your "Accept" recommendation.

Reviewer #1: All comments have been addressed

Reviewer #2: (No Response)

2. Is the manuscript technically sound, and do the data support the conclusions?

Reviewer #1: Yes

Reviewer #2: Yes

3. Has the statistical analysis been performed appropriately and rigorously? 

Reviewer #1: Yes

Reviewer #2: Yes

4. Have the authors made all data underlying the findings in their manuscript fully available?

Reviewer #1: Yes

Reviewer #2: Yes

5. Is the manuscript presented in an intelligible fashion and written in standard English?

Reviewer #1: Yes

Reviewer #2: Yes

6. Review Comments to the Author

Reviewer #1: (No Response)

Reviewer #2: Well done for the clarifications.

There is one remaining - point 7.

Typically if the endpoint name is relapse-free survival, it implies the events are relapse and death (the survival element of relapse free survival)

If relapse is the only event, it would typically be termed free-from-relapse. For clarity I would prefer this term to be used if this was the definition used.

7. PLOS authors have the option to publish the peer review history of their article (what does this mean?). If published, this will include your full peer review and any attached files.

Reviewer #1: No

Reviewer #2: No

---

## [Author Response · Author response to Decision Letter 2]

7 Jul 2022

Rebuttal letter 

Journal Requirements:

Rebuttal: Two duplicated references (22, 23) were removed. The final reference list includes 24 references (down from 26). No retracted articles could be found among the references. 

Reviewer #2:

Well done for the clarifications.

There is one remaining - point 7.

Typically if the endpoint name is relapse-free survival, it implies the events are relapse and death (the survival element of relapse free survival)

If relapse is the only event, it would typically be termed free-from-relapse. For clarity I would prefer this term to be used if this was the definition used.

Rebuttal: In the manuscript, relapse free survival (RFS) was defined as time from primary operation (i.e., initial treatment) until relapse (i.e., event) or end of follow-up (i.e., censor). Relapses were considered events, defined as increase in thyroglobulin level or histologically confirmation during follow-up. Deaths from other causes than PTC were considered censors in RFS. In the patient material, we saw two deaths from PTC. 

For the definition we refer to National Cancer Institute’s dictionary that defines RFS as “in cancer, the length of time after primary treatment for a cancer ends that the patient survives without any signs or symptoms of that cancer.” Therefore, we thank the reviewer for the comment but respectfully wish to keep the definition in the manuscript as is.

---

## [Decision Letter · Decision Letter 3]

26 Jul 2022

A deep learning–based algorithm for tall cell detection in papillary thyroid carcinoma

PONE-D-21-21384R3

Dear Dr. Stenman,

We’re pleased to inform you that your manuscript has been judged scientifically suitable for publication and will be formally accepted for publication once it meets all outstanding technical requirements.

Kind regards,

Alvaro Galli

Academic Editor

PLOS ONE

Additional Editor Comments (optional):

Reviewers' comments:

Reviewer's Responses to Questions

**Comments to the Author**

1. If the authors have adequately addressed your comments raised in a previous round of review and you feel that this manuscript is now acceptable for publication, you may indicate that here to bypass the “Comments to the Author” section, enter your conflict of interest statement in the “Confidential to Editor” section, and submit your "Accept" recommendation.

Reviewer #1: All comments have been addressed

2. Is the manuscript technically sound, and do the data support the conclusions?

Reviewer #1: (No Response)

3. Has the statistical analysis been performed appropriately and rigorously? 

Reviewer #1: (No Response)

4. Have the authors made all data underlying the findings in their manuscript fully available?

Reviewer #1: (No Response)

5. Is the manuscript presented in an intelligible fashion and written in standard English?

Reviewer #1: (No Response)

6. Review Comments to the Author

Reviewer #1: (No Response)

7. PLOS authors have the option to publish the peer review history of their article (what does this mean?). If published, this will include your full peer review and any attached files.

Reviewer #1: No

---

## [Editor Report · Acceptance letter]

1 Aug 2022

PONE-D-21-21384R3 

A deep learning–based algorithm for tall cell detection in papillary thyroid carcinoma 

Dear Dr. Stenman:

I'm pleased to inform you that your manuscript has been deemed suitable for publication in PLOS ONE. Congratulations! Your manuscript is now with our production department. 

Kind regards, 

on behalf of

Dr. Alvaro Galli 

Academic Editor

PLOS ONE